# Robust early-learning: Hindering the memorization of noisy labels

**Xiaobo Xia**[1]   **Tongliang Liu**[1][†]   **Bo Han**[2]
**Chen Gong**[3]   **Nannan Wang**[4]   **Zongyuan Ge**[5,6]   **Yi Chang**[7]
[1]Trustworthy Machine Learning Lab, School of Computer Science, The University of Sydney
[2]Department of Computer Science, Hong Kong Baptist University
[3]School of Computer Science and Engineering, Nanjing University of Science and Technology
[4]ISN State Key Laboratory, School of Telecommunications Engineering, Xidian University
[5]Medical AI Group, Faculty of Engineering, Monash University
[6]Airdoc Research, Monash University
[7]School of Artificial Intelligence, Jilin University

## Abstract

The *memorization effects* of deep networks show that they will first memorize training data with clean labels and then those with noisy labels. The *early stopping* method therefore can be exploited for learning with noisy labels. However, the side effect brought by noisy labels will influence the memorization of clean labels before early stopping. In this paper, motivated by the *lottery ticket hypothesis* which shows that only partial parameters are important for generalization, we find that only partial parameters are important for fitting clean labels and generalize well, which we term as *critical parameters*; while the other parameters tend to fit noisy labels and cannot generalize well, which we term as *non-critical parameters*. Based on this, we propose *robust early-learning* to reduce the side effect of noisy labels before early stopping and thus enhance the memorization of clean labels. Specifically, in each iteration, we divide all parameters into the critical and non-critical ones, and then perform different update rules for different types of parameters. Extensive experiments on benchmark-simulated and real-world label-noise datasets demonstrate the superiority of the proposed method over the state-of-the-art label-noise learning methods.

## 1 Introduction

Deep neural networks have achieved a remarkable success in various tasks, such as image classification (He et al., 2015), object detection (Ren et al., 2015), speech recognition (Graves et al., 2013), and machine translation (Wu et al., 2016). However, the success is largely attributed to large amounts of data with high-quality annotations, which is expensive or even infeasible in practice (Han et al., 2018a; Li et al., 2020a; Wu et al., 2020). On the other hand, many large-scale datasets are collected from image search engines or web crawlers, which inevitably involves noisy labels (Xiao et al., 2015; Li et al., 2017a; Zhu et al., 2021). As deep networks have large learning capacities and strong memorization power, they will ultimately overfit noisy labels, leading to poor generalization performance (Jiang et al., 2018; Nguyen et al., 2020). General regularization techniques such as dropout and weight decay cannot address this issue well (Zhang et al., 2017).

Fortunately, even though deep networks will fit all the labels eventually, they first fit data with clean labels, which helps generalization (Arpit et al., 2017; Han et al., 2018b; Yu et al., 2019; Liu et al., 2020). Thus, the *early stopping* method can be used to reduce overfitting to the noisy labels (Rolnick et al., 2017; Li et al., 2020b; Hu et al., 2020). However, the existence of noisy labels will still adversely affect the memorization of clean labels even in the early training stage. This will hurt generalization (Han et al., 2020). Intuitively, if we can reduce the side effect of noisy labels before early stopping, the generalization and robustness of the networks can be improved.

---

[†]Correspondence to Tongliang Liu (tongliang.liu@sydney.edu.au).

Note that *over-parameterization* of deep networks is one of the main reasons for overfitting to noisy labels (Zhang et al., 2017; Yao et al., 2020a). The *lottery ticket hypothesis* (Frankle & Carbin, 2018) shows that only partial parameters are important for generalization. The deep networks with these important parameters can generalize well, or even better by avoid overfitting. Motivated by this, for learning with noisy labels, it remains a question if we can divide the parameters into two parts to reduce the side effect brought by noisy labels, which enhances the memorization of clean labels and further improves the generalization performance of the deep networks.

In this paper, we present a novel and effective method to find which parameters are important for fitting data with clean labels, and which parameters tend to fit data with noisy labels. We term the former as *critical parameters*, and the latter as *non-critical parameters*. Then on this basis, we proposed *robust early-learning* to reduce the side effect of noisy labels before early stopping. Specifically, in each iteration during training, we first categorize all parameters into two parts, i.e., the critical parameters and the non-critical parameters. Then we designed different update rules for different types of parameters. For the critical ones, we perform *robust positive update*. This part of the parameters are updated using the gradients derived from the objective function and weight decay. For the non-critical ones, we perform *negative update*. Their values are penalized with the weight decay, and without the gradients derived from the objective function. Note that the gradients for updating are based on the loss between the prediction of deep networks and given labels. For the critical ones, they tend to fit data with clean (correct) labels to help generalization. Their gradients can therefore be exploited to update parameters. However, for the non-critical ones, they tend to fit data with noisy (incorrect) labels, which hurts generalization. Their gradients will misguide the deep networks to overfit data with noisy labels. Thus, we only use a regularization item, i.e., the weight decay, to update them. The weight decay will penalize their values to be zero, which means that they are penalized to be deactivated, and not to contribute to the generalization of deep networks. In this way, we can reduce the side effect of noisy labels and enhance the memorization of clean labels. In summary, the main contributions of this work are as follows:

• We propose a novel and effective method which can categorize the parameters into two parts according to whether they are important to fit data with clean labels.
• Different update rules have been designed for different types of the parameters to reduce the side effect of noisy labels before early stopping.
• We experimentally validate the proposed method on both synthetic noisy datasets and real-world noisy datasets, on which it achieves superior robustness compared with the state-of-the-art methods for learning with noisy labels.

**Related Work.** Early stopping is quite simple but effective in practice. It was used in supervised learning early (Prechelt, 1998; Caruana et al., 2001; Zhang et al., 2005; Yao et al., 2007). With the help of a validation set, training is then stopped before convergence to avoid the overfitting. While learning with noisy labels, the networks fit the data with clean labels before starting to overfit the data with noisy labels (Arpit et al., 2017). Early stopping is then formally proved to be valid for relieving overfitting to noisy labels (Rolnick et al., 2017; Li et al., 2020b). It has also been widely used in existing methods to improve robustness and generalization (Yu et al., 2018b; Xu et al., 2019; Yao et al., 2020b; Cheng et al., 2021).

The lottery ticket hypothesis (Frankle & Carbin, 2018) shows that deep networks are likely to be over-parameterized, and only partial parameters are important for generalization. With this part of the parameters, the small and sparsified networks can be trained to generalize well. While this work is motivated by the lottery ticket hypotheis, this work is fundamentally different from it. The lottery ticket hypothesis focuses on network compression. It aims to find a sparsified sub-network which has competitive generalization performance compared with the original network. This paper focuses on learning with noisy labels. We want to find the critical/non-critical parameters to reduce the side effect of noisy labels, which greatly improves the generalization performance.

Lots of work proposed various methods for training with noisy labels, such as exploiting a noise transition matrix (Liu & Tao, 2016; Hendrycks et al., 2018; Xia et al., 2020a; Li et al., 2021), using graph models (Xiao et al., 2015; Li et al., 2017b), using surrogate loss functions (Zhang & Sabuncu, 2018; Wang et al., 2019; Ma et al., 2020), meta-learning (Ren et al., 2018; Shu et al., 2020), and employing the small loss trick (Jiang et al., 2018; Han et al., 2018b; Yu et al., 2019). Some methods among them employ early stopping explicitly or implicitly (Patrini et al., 2017; Xia et al., 2019). We

also use early stopping in this paper. We are the first to hinder the memorization of noisy labels with analyzing the criticality of parameters.

**Organization.** The rest of the paper is organized as follows. In Section 2, we setup the problem and introduce the neural network optimization method. In Section 3, we discuss how to find the critical parameters and perform different update rules. In Section 4, we provide empirical evaluations of the proposed learning algorithm. Finally, Section 5 concludes the paper.

## 2 PRELIMINARIES

**Notation.** Vectors and matrices are denoted by bold-faced letters. The standard inner product between two vectors is denoted by $\langle \cdot, \cdot \rangle$. We use $\| \cdot \|_p$ as the $\ell_p$ norm of vectors or matrices. For a function $f$, we use $\nabla f$ to denote its gradient. Let $[n] = \{1, 2, \ldots, n\}$.

**Problem Setup.** Consider a classification task, there are $c$ classes. Let $\mathcal{X}$ and $\mathcal{Y}$ be the feature and label spaces respectively, where $\mathcal{X} \in \mathbb{R}^d$ with $d$ being the dimensionality, and $\mathcal{Y} = [c]$. The joint probability distribution over $\mathcal{X} \times \mathcal{Y}$ is denoted by $D$. Let $S = \{(\mathbf{x}_i, y_i)\}_{i=1}^n$ be an i.i.d. sample drawn from $D$, where $n$ denotes the sample size. In traditional supervised learning, by employing $S$, the aim is to learn a classifier that can assign labels precisely for given instances. While learning with noisy labels, we are given a sample with noisy labels $\widetilde{S} = \{(\mathbf{x}_i, \widetilde{y}_i)\}_{i=1}^n$, which is drawn from a corrupted joint probability distribution $\widetilde{D}$ rather than $D$. Here, $\widetilde{y}$ is the possibly corrupted label of the underlying clean label $y$. The aim is changed to learn a robust classifier that could assign clean labels to test data by only exploiting a training sample with noisy labels.

### 2.1 NEURAL NETWORK OPTIMIZATION METHOD

The optimization method is essential for training neural networks. Stochastic gradient descent (SGD) is the most popular one nowadays among the optimization methods (Allen-Zhu et al., 2019; Cao & Gu, 2019; Zou et al., 2020). Our proposed method is directly related to SGD. We analyze the optimization problem of typical supervised learning with clean labels as knowledge background. Consider a classifier to be trained, let $\mathcal{W} \in \mathbb{R}^m$ be all the parameters, where $m$ is the total number of the parameters. Let $L : \mathbb{R}^c \times \mathcal{Y} \to \mathbb{R}_+$ be the surrogate loss function, e.g., *cross entropy loss*. With a regularization item, e.g., $\ell_1$ regularizer, optimization method would involve minimizing an objective function as:

$$\min L(\mathcal{W}; S) = \min \frac{1}{n} \sum_{i=1}^n L(\mathcal{W}; (\mathbf{x}_i, y_i)) + \lambda \|\mathcal{W}\|_1, \tag{1}$$

where $\lambda \in \mathbb{R}_+$ is a regularization parameter. The update rules of the parameters $\mathcal{W}$ can be represented by the following formula:

$$\mathcal{W}(k+1) \leftarrow \mathcal{W}(k) - \eta \left( \frac{\partial L(\mathcal{W}(k); S^\star)}{\partial \mathcal{W}(k)} + \lambda \mathrm{sgn}(\mathcal{W}(k)) \right), \tag{2}$$

where $\eta > 0$ is the learning rate, $\mathcal{W}(k)$ is the set of the parameters at the $k$-th iteration, $\mathrm{sgn}(\cdot)$ is the standard sgn function in mathematics, and $S^\star$ is a subset randomly sampled from $S$. With SGD, the regularization parameter $\lambda$ is equivalent to the weight decay coefficient in the training process (Loshchilov & Hutter, 2019).

## 3 METHODOLOGY

In this section, we first introduce an alternative interpretation for the optimality criterion (Section 3.1). Then, we present how to determine the critical/non-critical parameters by exploiting this interpretation during training and the memorization effect of deep neural networks (Section 3.2). Finally, different update rules are proposed for different types of parameters to cope with noisy labels (Section 3.3).

### 3.1 OPTIMALITY CRITERION

For the optimization of the objective function $L(\mathcal{W}; S)$, the optimality will be achieved at $\mathcal{W}$ when $\nabla L(\mathcal{W}; S) = \mathbf{0}$ (Boyd et al., 2004; Bubeck, 2014). However, modern neural networks are complex and over-parameterized, which makes $\nabla L(\mathcal{W}; S)$ extremely high-dimensional. It is unintuitive for us to analyze high-dimensional vectors. The optimality is hard to be effectively judged. To address this issue, we will use a more intuitive interpretation for optimality criterion in this paper, which can associate the optimization with a scalar. Specifically, if we let $G(t) = L(t\mathcal{W}; S)$,

$G'(t) = \nabla L(t\mathcal{W}; S)^\top \mathcal{W}$. Let $t = 1$, then $G'(1) = \nabla L(\mathcal{W}; S)^\top \mathcal{W} = \langle \nabla L(\mathcal{W}; S), \mathcal{W} \rangle$. We know that the optimality can be reached when $\nabla L(\mathcal{W}; S) = \mathbf{0}$, then $G'(1) = 0$. In this way, the optimality can be checked by exploiting the scalar $G'(1)$. Note that the new optimality criterion is sufficient, but is not necessary. In this paper, we focus on learning with noisy labels, and the necessity of the new optimality criterion does not affect the effectiveness of the proposed method. We will discuss this carefully in the next subsection.

### 3.2 Judging the importance of network parameters

We have shown that the optimization of the objective function can be related to a scalar $G'(1)$. Its value is equal to the *inner product* between the value of the parameters and the gradient w.r.t. the parameters. To achieve the optimality, we need to push the value of $G'(1)$ to be zero. Thus, we can judge the importance of each parameter through its influence on the value of $G'(1)$. Note that the memorization effects of deep networks show they first memorize the data with clean labels. The parameters that contribute to the optmiality at the early stage are therefore important for clean labels. Consider a parameter, denoted by $w_i \in \mathcal{W}$, its gradient is $\nabla L(w_i; S)$. The judgement criteria is denoted by $g_i$, i.e.,

$$g_i = |\nabla L(w_i; S) \times w_i|, i \in [m]. \tag{3}$$

If the value of $g_i$ is large, $w_i$ is viewed as a *critical parameter*, as $g_i$ has a great influence on the value of $G'(1)$. On the contrary, if the value of $g_i$ is small, e.g., zero or very close to zero, $w_i$ is regarded to be a *non-critical parameter*. It is not important for fitting clean labels. If we update it, it will tend to fit noisy labels.

The underlying issue of directly using the gradient of $L(w_i; S)$ as a criterion for the criticality can be identified. When we only exploit gradient information, we ignore the value of the parameter $w_i$. However, if the value is zero or close to zero, the parameter is *inactivated*. It is also non-critical for optimality (Han et al., 2015; Frankle & Carbin, 2018; Lee et al., 2019). Note that we use early stopping in this paper. The deep networks mainly fit clean labels in the early training. Thus, even with the existence of noisy labels, we can use the criterion to analyze the criticality of the parameters.

It should be noted when $g_i = 0$, there will be three possible scenarios: (1) only $\nabla L(w_i; S) = 0$; (2) only $w_i = 0$; (3) both $\nabla L(w_i; S) = 0$ and $w_i = 0$. In all three cases, we can judge the importance of parameter $w_i$ using $g_i$ as analyzed. In other words, when $g_i = 0$, we allow that the value of $\nabla L(w_i; S)$ is not zero, i.e., the new optimality criterion is not necessary, which does not influence the effectiveness of the proposed method.

### 3.3 Combating noisy labels with different update rules

We have presented how to judge the importance of the parameters, and then divide them into the critical ones and non-critical ones. We exploit the label noise rate to help divide the parameters into the critical/non-critical ones. Intuitively, if the noise rate is high, the number of clean labels is small. The number of required critical parameters for memorizing clean labels is then small. The number of the critical parameters has a negative correlation with noise rate. We therefore use the noise rate to help identify the critical parameters. We use $\tau$ to denote the noise rate. If $\tau$ is not known in advanced, it can be easily inferred (Liu & Tao, 2016; Yu et al., 2018a). We show that the proposed method is insensitive to the estimation result of the noise rate in Section 4.4. Then, the number of the critical parameters can be defined as:

$$m_c = (1 - \tau)m. \tag{4}$$

In each iteration, for each parameter $w_i$, $i \in [m]$. The critical and non-critical parameters are determined according to the result of numerical sorting of $g_i$, which has been explained before. The critical and non-critical parameters are denoted by $\mathcal{W}_c$ and $\mathcal{W}_n$ respectively. Two different update strategies are performed for two types of the parameters.

**Robust positive update**. For the critical ones $\mathcal{W}_c$, we use the gradients derived from the objective function and weight decay. We clip the gradients to perform *gradient decay* in this paper. The update rule is:

$$\mathcal{W}_c(k + 1) \leftarrow \mathcal{W}_c(k) - \eta \left( (1 - \tau) \frac{\partial L(\mathcal{W}_c(k); \widetilde{S}^\star)}{\partial \mathcal{W}_c(k)} + \lambda \mathrm{sgn}(\mathcal{W}_c(k)) \right), \tag{5}$$

where $\widetilde{S}^\star$ is a subset randomly sampled from $\widetilde{S}$. Note that we directly use $\widetilde{S}^\star$ here, rather than $S^\star$ in Eq. (2). It is because the proposed method exploits the memorization effects of deep networks.

---

**Algorithm 1** CDR algorithm.

---

1: **Input**: initialization parameters $\mathcal{W}$, noisy training set $\mathcal{D}_t$, noisy validation set $\mathcal{D}_v$, learning rate $\eta$, weight decay coefficient $\lambda$, fixed $\tau$, epoch $T$ and $T_{\max}$, iteration $N_{\max}$;

**for** $T = 1, 2, \ldots, T_{\max}$ **do**
   2: **Shuffle** training set $\mathcal{D}_t$;
   **for** $N = 1, \ldots, N_{\max}$ **do**
      3: **Fetch** mini-batch $\bar{\mathcal{D}}_t$ from $\mathcal{D}_t$;
      4: **Divide** $\mathcal{W}$ into $\mathcal{W}_c$ and $\mathcal{W}_n$ with Eq. (3) and Eq. (4); //define the types of the parameters;
      5: **Update** $\mathcal{W}_c$ with Eq. (5);            //update $\mathcal{W}_c$ using the robust positive update;
      6: **Update** $\mathcal{W}_n$ with Eq. (6);            //update $\mathcal{W}_n$ using the negative update;
   **end**
**end**
//Early stopping criterion: if the minimum classification error is achieved with $\mathcal{W}$ on $\mathcal{D}_v$
8: **Output**: parameters $\mathcal{W}$ after update.

---

Though we have only noisy training data, deep networks will first memorize training data with clean labels. As can be seen in Eq. (5), the gradient decay coefficient is set to $1 - \tau$, which can prevent over-confident descent steps in the training process.

**Negative update**. For the non-critical parameters $\mathcal{W}_n$, we only use the weight decay to update them. The update rule is:

$$\mathcal{W}_n(k + 1) \leftarrow \mathcal{W}_n(k) - \eta\lambda\mathrm{sgn}(\mathcal{W}_n(k)). \tag{6}$$

The gradients of the objective function exploit loss between the prediction of deep networks and given labels. Robust positive update uses the gradients to update the critical ones, which helps deep networks memorize clean labels. For the non-critical ones, they tend to overfit noisy labels, their gradients are misleading for generalization. Thus, we only use the weight decay, to update them. The weight decay will penalize their values to be zero and help generalization (Arora et al., 2018). As they are deactivated, they will not contribute to the memorization or generalization. The use of two update rules makes us achieve the goal, i.e., reducing the side effect of noisy labels and thus enhance the memorization of clean labels. The overall procedure of combating noisy labels with different update rules (CDR) is summarized in Algorithm 1.

## 4 EXPERIMENTS

In this section, we first introduce the datasets used, and implementation details in the experiments (Section 4.1). We next introduce the methods used for comparison in this paper (Section 4.2). The ablation study is conducted to show that the proposed method is not sensitive to the estimation result of the noise rate (Section 4.3). Finally, we present the experimental results on synthetic and real-world noisy datasets to show the effectiveness of the proposed method (Section 4.4).

### 4.1 DATASETS AND IMPLEMENTATION DETAILS

To verify the effectiveness of the proposed method, we run experiments on the manually corrupted version of four datasets, i.e., *MNIST* (LeCun et al., 1998), *F-MNIST* (Xiao et al., 2017), *CIFAR-10* and *CIFAR-100* (Krizhevsky & Hinton, 2009), and two real-world noisy datasets, i.e., *Food-101* (Bossard et al., 2014) and *WebVision* (Li et al., 2017a). *MNIST* and *F-MNIST* both have 28×28 gayscale images of 10 classes including 60,000 training images and 10,000 test images. *CIFAR-10* and *CIFAR-100* both have $32 \times 32 \times 3$ color images including 50,000 training images and 10,000 test images. *CIFAR-10* has 10 classes while *CIFAR-100* has 100 classes. *Food-101* consists of 101 food categories, with 101,000 images. For each class, 250 manually reviewed clean test images are provided as well as 750 training images with real-world label noise. *WebVision* contains 2.4 million images crawled from the websites using the 1,000 concepts in *ImageNet ILSVRC12* (Deng et al., 2009). Following the "Mini" setting in (Jiang et al., 2018; Chen et al., 2019; Ma et al., 2020), we take the first 50 classes of the Google resized image subset, and evaluate the trained networks on the same 50 classes of the *ILSVRC12* validation set, which is exploited as a test set. For all datasets, following prior works (Patrini et al., 2017; Wang et al., 2021b), we leave out 10% training data as a validation set, which is for early stopping.

We consider four types of synthetic label noise in this paper, i.e., symmetric noise, asymmetric noise, pairflip noise and instance-dependent noise (abbreviated as instance noise). These settings are widely used in existing works (Ma et al., 2018; Thulasidasan et al., 2019; Pleiss et al., 2020; Wang et al., 2021a). The noise rates $\tau$ are set to 20% and 40%. The details of the noise setting are described as follows:

• Symmetric noise: this kind of label noise is generated by flipping labels in each class uniformly to incorrect labels of other classes.
• Asymmetric noise: this kind of label noise is generated by flipping labels within a set of similar classes. In this paper, for *MNIST*, flipping $2 \rightarrow 7$, $3 \rightarrow 8$, $5 \leftrightarrow 6$. For *F-MNIST*, flipping T-SHIRT → SHIRT, PULLOVER → COAT, SANDALS → SNEAKER. For *CIFAR-10*, TRUCK → AUTOMOBILE, BIRD → AIRPLANE, DEER → HORSE, CAT↔DOG. For *CIFAR-100*, the 100 classes are grouped into 20 super-classes, and each has 5 sub-classes. Each class is then flipped into the next within the same super-class.
• Pairflip noise: the noise flips each class to its adjacent class. More explanation about this noise setting can be found in (Yu et al., 2019; Zheng et al., 2020; Lyu & Tsang, 2020).
• Instance noise: the noise is quite realistic, where the probability that an instance is mislabeled depends on its features. Following (Xia et al., 2020b), we generate this type of label noise to validate the effectiveness of the proposed method.

For *MNIST*, we train a LeNet (LeCun et al., 1998) with batch size 32. For *F-MNIST*, we train a ResNet-50 (He et al., 2015) with batch size 32. For *CIFAR-10* and *CIFAR-100*, we train a ResNet-50 with batch size 64, and typical data augmentations including random crop and horizontal flip are applied. For all the training, we use SGD optimizer with momentum 0.9 and weight decay is set to $10^{-3}$. The initial learning rate is set to $10^{-2}$. For *Food-101*, we use a ResNet-50 pre-trained on ImageNet with batch size 32. The initial learning rate is changed to $10^{-3}$. For *WebVision*, we use an Inception-ResNet v2 (Szegedy et al., 2016) with batch size 128. The initial learning rate is set to $10^{-1}$. We set 100 epochs in total for all the experiments. For fair comparison, all the codes are implemented in PyTorch 1.2.0 with CUDA 10.0, and run on NVIDIA Tesla V100 GPUs. Our implementation is available at https://github.com/xiaoboxia/CDR.

## 4.2 COMPARISON METHODS

We compare the proposed method with the following methods: (1) CE, which trains the deep neural networks with the cross entropy loss on noisy datasets. (2) GCE (Zhang & Sabuncu, 2018), which unites the mean absolute error loss and the cross entropy loss to handle noisy labels. The hyperparameter $q$ in this work is set to 0.7. (3) DMI (Xu et al., 2019), which copes with noisy labels from the perspective of information theory. (4) APL (Ma et al., 2020), which combines two mutually reinforcing robust loss functions. For this baseline, we employ its combination of NCE and RCE for a comparison. (5) MentorNet (Jiang et al., 2018), which learns a curriculum to filter out noisy data. (6) Co-teaching (Han et al., 2018b), which maintains two networks and cross-trains on the instances with small loss. (7) Co-teaching+ (Yu et al., 2019), which maintains two networks and finds small loss instances among the prediction disagreement data for training. (8) S2E (Yao et al., 2020a), which exploits automated machine learning to handle noisy labels. (9) Forward (Patrini et al., 2017), which estimates the noise transition matrix to correct the training loss. (10) T-Revision (Xia et al., 2019), which employs importance reweighting technique and introduces a slack variable to revise the noise transition matrix. (11) Joint (Tanaka et al., 2018), which jointly optimizes the network parameters and the sample labels. The hyperparameters $\alpha$ and $\beta$ are set to 1.2 and 0.8 respectively. Note that we do not compare with some state-of-the-art methods like SELF (Nguyen et al., 2020) and DivideMix (Li et al., 2020a) as baselines because of the following reasons. (1) Their proposed methods are aggregations of multiple techniques while this paper only focuses on one, therefore the comparison is not fair. (2) We are focusing on proving the concept, i.e., how to reduce the side effect of noisy labels before early stopping, but not on boosting the classification performance.

## 4.3 ABLATION STUDY

We need the noise rate to determine the number of different types of the parameters and the gradient decay coefficient as mentioned above. Compared with symmetric noise, asymmetric noise and pairflip noise, the noise rate of instance noise is hard to be estimated (Cheng et al., 2020; Xia et al., 2020b). We present that our proposed method is insensitive to the estimation result of noise rate. The experiments are conducted on *CIFAR-10* and *CIFAR-100* datasets with instance noise. The noise rates are set to 20% and 40%, respectively. In Figure 1, we show that how the classificaition performance

Table 1: Mean and standard deviations of classification accuracy (percentage) on synthetic noisy datasets with different noise levels. The experimental results are reported over five trials. The best mean results are bolded.

| Dataset | Method | Symmetric | | Asymmetric | | Pairflip | | Instance | |
|---|---|---|---|---|---|---|---|---|---|
| | | 20% | 40% | 20% | 40% | 20% | 40% | 20% | 40% |
| MNIST | CE | 98.60±0.07 | 98.18±0.16 | 99.00±0.08 | 98.31±0.28 | 98.74±0.17 | 94.08±0.93 | 98.14±0.02 | 92.76±0.21 |
| | GCE | 98.84±0.12 | 98.12±0.33 | 98.92±0.09 | 98.31±0.27 | 98.94±0.05 | 97.39±0.62 | 98.17±0.08 | 94.97±0.32 |
| | DMI | 98.94±0.02 | 98.62±0.17 | 99.07±0.10 | 98.61±0.26 | 98.96±0.12 | 97.27±0.39 | 98.34±0.17 | 95.34±0.28 |
| | APL | 98.74±0.09 | 97.04±0.35 | 98.90±0.07 | 97.23±0.73 | 98.24±0.39 | 95.24±1.49 | 97.03±0.29 | 90.04±3.93 |
| | MentorNet | 97.21±0.13 | 93.96±0.76 | 98.51±0.09 | 93.47±0.80 | 97.25±0.32 | 93.27±0.73 | 95.17±0.26 | 90.05±1.43 |
| | Co-teaching | 97.22±0.18 | 94.64±0.33 | 98.63±0.12 | 93.62±1.27 | 97.44±0.26 | 94.81±0.45 | 97.32±0.15 | 92.45±0.59 |
| | Co-teaching+ | 98.11±0.07 | 95.87±0.27 | 98.83±0.08 | 96.65±1.73 | 98.81±0.12 | 95.42±0.33 | 98.07±0.12 | 94.37±0.48 |
| | S2E | 98.93±0.39 | 93.23±2.37 | 99.23±0.07 | 98.31±0.13 | 99.10±0.04 | 80.15±3.78 | 98.42±0.47 | 83.38±0.94 |
| | Forward | 98.10±0.12 | 96.83±0.28 | 98.79±0.28 | 97.94±0.47 | 98.62±0.16 | 95.37±0.70 | 97.87±0.21 | 92.30±0.18 |
| | T-Revision | 98.93±0.07 | 98.40±0.53 | 99.05±0.16 | 98.23±0.54 | 98.82±0.07 | 97.43±0.19 | 98.33±0.15 | **95.64±0.34** |
| | Joint | 98.54±0.13 | 98.30±0.28 | 98.96±0.05 | 98.40±0.11 | 98.70±0.08 | 96.33±0.82 | 98.11±0.13 | 93.15±0.43 |
| | CDR | **99.00±0.04** | **98.80±0.12** | **99.30±0.04** | **98.80±0.17** | **99.17±0.08** | **98.12±0.23** | **98.49±0.09** | 94.45±1.04 |
| F-MNIST | CE | 90.36±0.21 | 87.61±0.72 | 91.31±0.13 | 86.43±2.01 | 91.65±0.12 | 76.42±4.13 | 88.81±0.67 | 78.62±2.92 |
| | GCE | 91.77±0.13 | 90.02±0.37 | 91.45±0.29 | 73.62±2.92 | 91.99±0.36 | 84.21±2.05 | 91.06±0.55 | 74.82±0.94 |
| | DMI | 91.87±0.26 | 88.65±0.37 | 92.33±0.11 | 89.62±0.48 | 91.33±0.37 | 83.93±0.92 | 90.87±0.15 | 80.51±0.66 |
| | APL | 87.23±0.19 | 73.62±0.88 | 86.21±0.24 | 81.03±3.09 | 84.52±0.73 | 76.39±2.85 | 84.38±0.70 | 60.38±8.37 |
| | MentorNet | 88.12±0.12 | 86.05±0.27 | 89.76±0.18 | 68.93±3.20 | 87.39±0.57 | 76.90±5.72 | 86.50±0.26 | 78.37±0.95 |
| | Co-teaching | 89.03±0.32 | 87.04±0.69 | 92.03±0.16 | 72.23±4.38 | 89.63±0.78 | 84.10±0.92 | 89.27±0.86 | 83.49±1.27 |
| | Co-teaching+ | 91.34±0.17 | 90.23±0.21 | 83.98±1.05 | 66.27±3.01 | 91.08±0.25 | 72.65±0.49 | 83.78±0.73 | 38.79±9.93 |
| | S2E | 90.89±0.27 | 75.68±3.73 | 91.20±0.31 | 87.06±0.50 | 91.52±0.19 | 72.09±2.15 | 89.17±0.32 | 72.62±2.73 |
| | Forward | 90.72±0.19 | 88.05±0.73 | 92.05±0.21 | 85.42±0.74 | 90.02±0.87 | 83.06±0.79 | 87.95±0.75 | 75.34±1.89 |
| | T-Revision | 91.95±0.20 | 90.35±0.28 | 92.07±0.11 | 88.53±0.32 | 91.06±0.19 | 85.67±0.88 | 91.05±0.28 | 84.34±1.37 |
| | Joint | 82.01±0.77 | 72.36±2.84 | 85.92±0.83 | 73.09±0.91 | 86.04±0.99 | 70.87±3.95 | 82.07±0.94 | 50.62±4.77 |
| | CDR | **92.24±0.11** | **90.91±0.27** | **93.01±0.14** | **90.37±0.32** | **93.06±0.19** | **87.55±1.07** | **91.52±0.17** | **85.04±1.02** |
| CIFAR-10 | CE | 89.14±0.41 | 86.25±1.32 | 88.21±0.19 | 86.37±1.03 | 89.68±0.72 | 86.53±0.37 | 86.73±0.36 | 75.33±2.72 |
| | GCE | 89.48±0.28 | 86.07±0.41 | 89.03±0.21 | 84.12±1.24 | 88.58±0.34 | 83.23±3.98 | 88.02±0.34 | 76.89±0.96 |
| | DMI | 89.29±0.30 | 86.89±1.07 | 89.37±0.82 | 86.32±1.17 | 88.41±1.01 | 84.02±1.73 | 88.93±0.29 | 79.35±2.17 |
| | APL | 88.21±0.32 | 81.07±1.36 | 89.03±0.75 | 85.10±2.42 | 87.34±1.44 | 80.12±3.65 | 76.31±2.24 | 50.73±4.89 |
| | MentorNet | 83.26±0.72 | 78.37±1.73 | 84.07±0.59 | 60.22±3.47 | 78.73±0.89 | 69.37±3.28 | 83.06±0.92 | 73.40±2.19 |
| | Co-teaching | 88.20±0.27 | 84.45±0.68 | 87.42±0.38 | 64.03±0.73 | 82.66±0.32 | 73.68±0.62 | 86.71±0.79 | 81.14±1.32 |
| | Co-teaching+ | 86.47±0.92 | 78.93±0.74 | 85.37±0.47 | 63.17±3.48 | 84.01±1.01 | 70.17±1.37 | 85.92±0.26 | 57.95±3.17 |
| | S2E | **90.26±0.24** | 75.20±2.05 | 90.73±0.32 | 87.83±0.97 | 89.92±0.37 | 76.18±1.93 | 90.32±0.21 | 68.93±1.86 |
| | Forward | 88.36±0.34 | 86.47±0.98 | 89.30±0.71 | 85.33±1.48 | 87.62±0.24 | 83.23±1.30 | 85.39±0.23 | 76.88±1.26 |
| | T-Revision | 89.43±0.62 | 86.98±0.87 | 89.94±0.74 | 88.11±1.22 | 91.01±0.29 | 87.10±1.38 | 90.43±0.38 | 85.46±1.04 |
| | Joint | 89.94±0.25 | 87.17±0.35 | 90.83±0.18 | 88.24±0.79 | 91.31±0.73 | 85.62±1.75 | 90.13±0.34 | 85.23±0.74 |
| | CDR | **90.26±0.31** | **87.19±0.43** | **92.00±0.27** | **88.68±0.67** | **92.11±0.23** | **88.58±0.39** | **91.14±0.23** | **86.25±0.57** |
| CIFAR-100 | CE | 63.93±0.72 | 56.82±0.82 | 64.12±0.54 | 52.86±0.92 | 64.10±0.46 | 52.77±0.79 | 63.33±0.29 | 50.84±0.89 |
| | GCE | 65.62±0.82 | 57.97±1.21 | 65.34±0.64 | 54.35±1.28 | 62.32±1.04 | 55.03±1.25 | 66.67±0.40 | 55.14±1.77 |
| | DMI | 62.77±0.92 | 57.42±0.53 | 64.30±0.84 | 51.31±2.73 | 58.77±0.64 | 42.89±0.77 | 59.04±0.35 | 46.99±0.62 |
| | APL | 59.37±0.82 | 51.03±1.04 | 54.31±0.84 | 48.22±1.35 | 59.77±0.74 | 53.25±0.92 | 49.17±2.72 | 38.18±4.04 |
| | MentorNet | 57.27±1.32 | 49.01±2.09 | 54.10±0.92 | 33.21±1.82 | 54.73±1.26 | 45.31±2.93 | 50.02±0.73 | 36.27±1.64 |
| | Co-teaching | 61.47±0.41 | 53.44±0.40 | 57.35±0.82 | 37.62±1.77 | 58.11±0.47 | 48.46±0.64 | 57.73±0.37 | 43.28±0.55 |
| | Co-teaching+ | 64.13±0.32 | 55.92±0.81 | 58.97±1.19 | 40.16±2.74 | 56.31±0.41 | 38.03±0.55 | 55.45±0.57 | 41.11±1.32 |
| | S2E | 64.21±0.72 | 43.12±2.77 | 63.92±0.46 | 42.45±1.73 | 58.21±0.43 | 41.74±2.09 | 61.08±0.59 | 47.06±1.93 |
| | Forward | 54.88±0.92 | 45.64±1.77 | 64.07±1.02 | 53.84±2.71 | 58.37±0.56 | 39.82±0.73 | 58.55±0.31 | 46.42±0.95 |
| | T-Revision | 64.67±0.38 | 57.15±1.02 | 68.02±0.53 | 54.93±1.09 | 62.69±0.73 | 52.31±1.46 | 60.22±0.68 | 50.23±1.79 |
| | Joint | 66.12±0.42 | 59.45±0.68 | 68.29±0.25 | 55.53±0.47 | 67.35±0.31 | 52.22±1.85 | 65.91±0.43 | 55.09±0.93 |
| | CDR | **68.68±0.33** | **62.72±0.38** | **70.64±0.51** | **55.58±0.78** | **71.93±0.57** | **56.94±1.30** | **69.82±0.42** | **61.03±0.77** |

of the proposed method varies with the change of the estimated noise rate. We can clearly see that the proposed method is robust to the estimation result of the noise rate.

In this paper, we set the proportion of non-critical parameters to the noise rate. Also, it is interesting to investigate that how the proposed method works if we set the proportion of non-critical parameters as a constant number. Note that if we set the constant number too randomly, the performance of the proposed method may be hurt. Therefore, we use a noisy validation set to locate it and compare the difference between the located constant and the noise rate. The search of the constant is within the range $\{0.10, 0.20, \dots, 0.90\}$. The experiments are conducted on *MNIST*, *F-MNIST*, and *CIFAR-10*. The experimental results are provided in Table 2. As we can see, in many cases, the located constant and the label noise rate are numerically equal. However, it is complicated to locate a suitable constant with a noisy validation set, as there is a huge search range. On the contrary, the noise rate always can be estimated effectively (Liu & Tao, 2016; Yu et al., 2018a). Therefore, it is reasonable and feasible to assume that the proportion of non-critical parameters is the same as the noise rate.

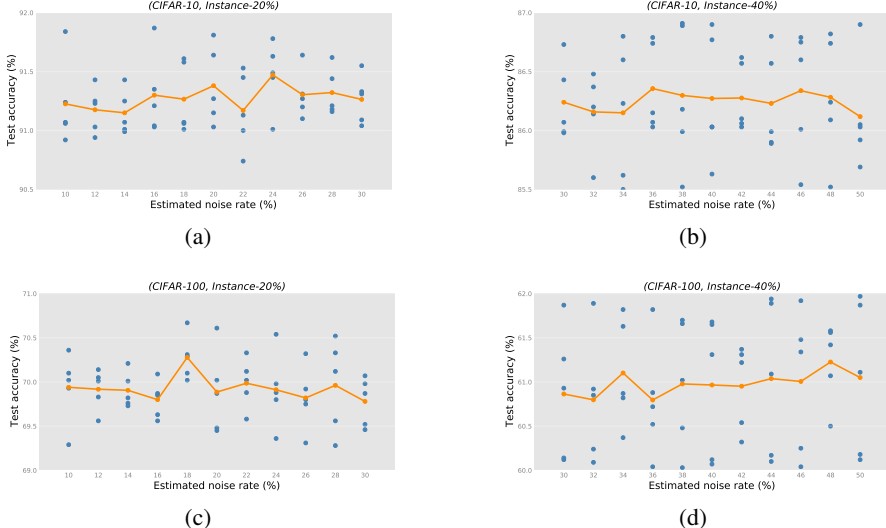

Figure 1: Illustration of robustness to the estimation result of noise rate. For each estimated noise rate, we report experimental results over five trials. The blue dots represent the result of each experiment. The orange dots represent the mean of five experimental results in each case.

Table 2: The located constant on synthetic noisy datasets with different noise levels. The result with an underline means that the located constant and the noise rate are numerically equal.

| Dataset | Symmetric | | Asymmetric | | Pairflip | | Instance | |
|---------|-----------|------|------------|------|----------|------|----------|------|
| | 20% | 40% | 20% | 40% | 20% | 40% | 20% | 40% |
| *MNIST* | 0.20 | 0.50 | 0.20 | 0.40 | 0.20 | 0.50 | 0.30 | 0.60 |
| *F-MNIST* | 0.20 | 0.40 | 0.20 | 0.50 | 0.30 | 0.50 | 0.20 | 0.60 |
| *CIFAR-10* | 0.30 | 0.50 | 0.30 | 0.30 | 0.30 | 0.50 | 0.20 | 0.40 |

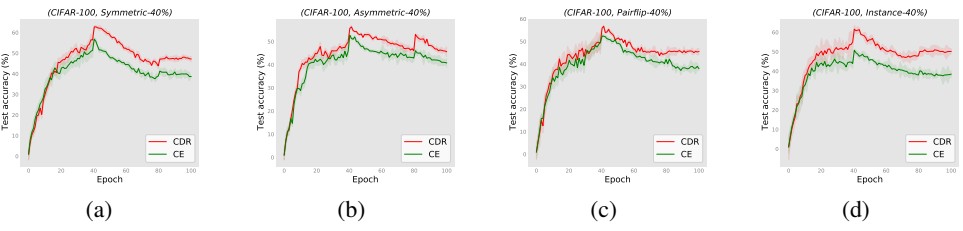

Figure 2: Illustration of the experimental results on noisy *CIFAR-100*. We can clearly see that the proposed method (CDR) can reduce the side effect of noisy labels at the early training stage, which improves generalization (red line *vs.* green line).

### 4.4 CLASSIFICATION PERFORMANCE ON NOISY DATASETS

**Results on synthetic noisy datasets.** Table 1 shows the experimental results on four synthetic noisy datasets with various types of noisy labels. For *MNIST*, as can be seen, our proposed method produce the best results in the vast majority of cases. When the noise is instance-dependent, the proposed method achieves competitive results. Note that T-Revision achieves the best classification performance in this case. Compared with the other synthetic noisy datasets, *MNIST* is less challenging. In this case, T-Revision thus can exploit the noise transition matrix and the slack variable to well model label noise, which leads to the best performance. However, for the instance-dependent label noise on the other datasets, i.e., *F-MNIST*, *CIFAR-10*, and *CIFAR-100*, estimating the transition matrices does not work well and the proposed robust eary-learning method achieves the best performance.

Table 3: Classification accuracy (percentage) on *Food-101* dataset. The best result is in bold.

| CE | GCE | DMI | APL | MentorNet | Co-teaching |
|---|---|---|---|---|---|
| 84.03 | 84.96 | 85.52 | 82.17 | 81.25 | 83.73 |
| Co-teaching+ | S2E | Forward | T-Revision | Joint | CDR |
| 76.89 | 84.97 | 85.52 | 85.97 | 83.10 | **86.36** |

Table 4: Top-1 validation accuracies (percentage) on clean *ILSVRC12* validation set of Inception-ResNet v2 models trained on *WebVision* dataset, under the "Mini" setting in (Jiang et al., 2018; Chen et al., 2019; Ma et al., 2020). The best result is in bold.

| CE | GCE | DMI | APL | MentorNet | Co-teaching |
|---|---|---|---|---|---|
| 57.34 | 55.62 | 56.93 | 61.27 | 57.66 | 61.22 |
| Co-teaching+ | S2E | Forward | T-Revision | Joint | CDR |
| 33.26 | 54.33 | 56.39 | 60.58 | 47.60 | **61.85** |

For *F-MNIST* and *CIFAR-10*, our proposed method is consistently superior to other state-of-the-art methods across all the settings. Note that S2E achieves impressive performance in the case of Symmetric-20% on *CIFAR-10*. However, it fails to generalize well compared with the proposed method, especially in the cases of 40% label noise rate. In contrast, CDR achieves a clear lead over S2E in these cases, which verifies the effectiveness of the proposed method. Lastly, for the more challenging dataset, i.e., *CIFAR-100*, the proposed method once again outperforms all the baseline methods. In particular, in the very challenging case of Instance-40%, the proposed method takes a nearly 6% lead compared to the second best method GCE.

**Results on real-world noisy datasets.** The experimental results on *Food-101* and *WebVision* datasets are reported in Table 3 and 4. Again, on classification accuracy, our method surpasses all other baselines. This verifies the effectiveness of our proposed method against real-world label noise.

Note that the CE method exploits early stopping in all experiments. Compared the classification performance of CE with the classification performance of CDR, i.e., early stopping *vs.* robust early stopping, we can clearly see that the proposed method can achieve better performance. We also present the illustration of the experimental results on *CIFAR-100* with 40% noise. As shown in Figure 2, CDR can effectively reduce the side effect of noisy labels at the early training stage. The illustrations of the experimental results on the other settings can be found in Appendix A.

## 5 CONCLUSION

In this paper, motivated by the lottery ticket hypothesis, we provide a novel method to distinguish the critical parameters and non-critical parameters for fitting clean labels. Then we propose different update rules for different types of parameters to reduce the side effect before early stopping. The proposed method is very effective for learning with noisy labels, which is supported by experiments on synthetic datasets with various types of label noise as well as on real-world datasets. Our method is simple and orthogonal to other methods. We believe that this opens up new possibilities in the topics of learning with noisy labels. It would be interesting to explore the potential characteristics of the updated parameters such as the distance to initial parameters (Li et al., 2020b; Hu et al., 2020) and the mutual information with the vector of all training labels given inputs (Harutyunyan et al., 2020).

## ACKNOWLEDGMENTS

TLL was supported by Australian Research Council Project DE-190101473 and DP-180103424. BH was supported by the RGC Early Career Scheme No. 22200720, NSFC Young Scientists Fund No. 62006202, HKBU Tier-1 Start-up Grant, and HKBU CSD Departmental Incentive Scheme. CG was supported by NSF of China (No. 61973162) and CCF-Tencent Open Fund (No: RAGR20200101). NNW was supported by National Key Research and Development Program of China under Grant 2018AAA0103202. ZYG was supported by the Airdoc-Monash research centre fellowship. YC was supported by the NSF of China (No.61976102, No.U19A2065). We thank anonymous reviewers for giving constructive comments.

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

# A  DETAILED EXPERIMENTAL RESULTS

In section 4.4, we provide the illustrations of the experimental results. However, because of limited space, we only show the illustrations of the experimental results on noisy *CIFAR-100*. The noise rate is set to 40%. In this supplementary material, we provide the illustrations of the experimental results on the other employed datasets and settings as follows.

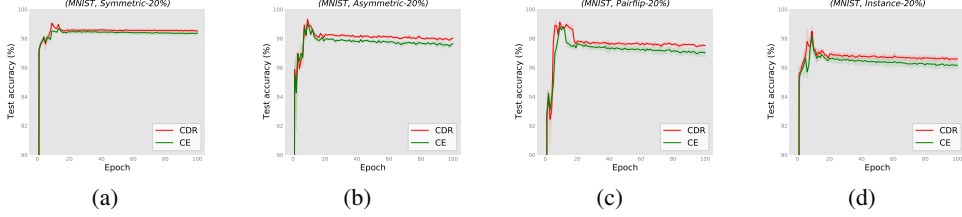

Figure 3: Illustration of the experimental results on noisy *MNIST*. The noise rate is set to 20%.

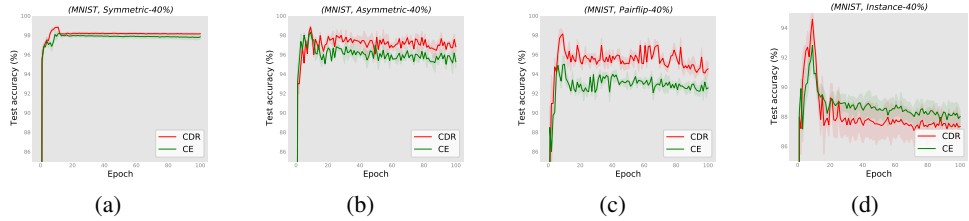

Figure 4: Illustration of the experimental results on noisy *MNIST*. The noise rate is set to 40%.

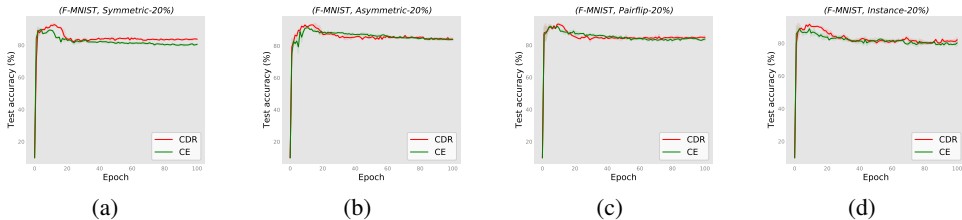

Figure 5: Illustration of the experimental results on noisy *F-MNIST*. The noise rate is set to 20%.

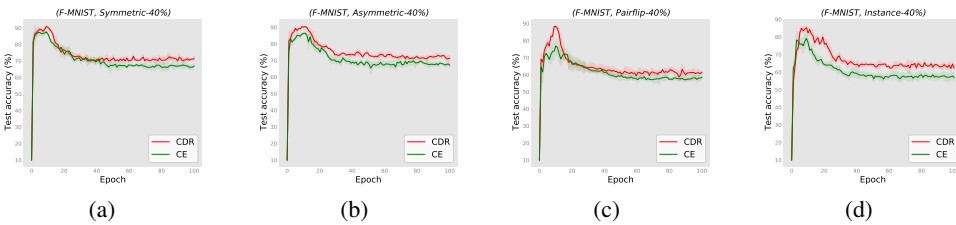

Figure 6: Illustration of the experimental results on noisy *F-MNIST*. The noise rate is set to 40%.

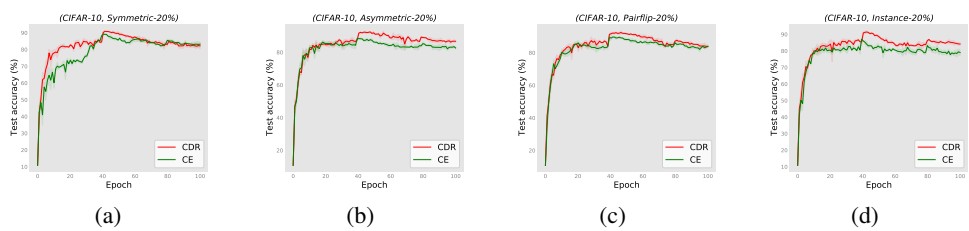

Figure 7: Illustration of the experimental results on noisy *CIFAR-10*. The noise rate is set to 20%.

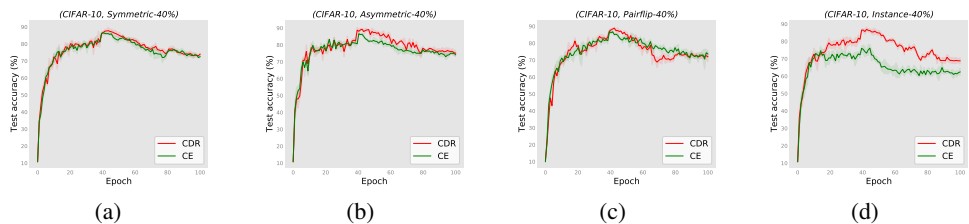

Figure 8: Illustration of the experimental results on noisy *CIFAR-10*. The noise rate is set to 40%.

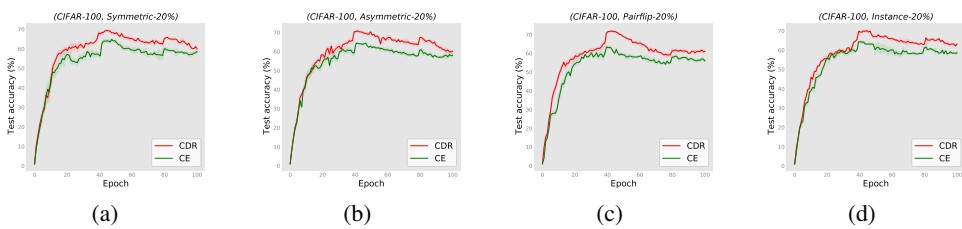

Figure 9: Illustration of the experimental results on noisy *CIFAR-100*. The noise rate is set to 20%.

