# OpenReview forum: "Robust early-learning: Hindering the memorization of noisy labels"
_ICLR.cc/2021/Conference — ICLR 2021 Poster_

### Official Review · AnonReviewer4 · 2020-10-20
**New technique for learning with noisy labels**

**Rating:** 6
**Confidence:** 4

**Review:**

This paper proposes a method for deep learning with noisy labels, which distinguishes the critical parameters and non-critical parameters for fitting clean labels and updates them by different rules. The method is easy to implement and the empirical results are promising. Experiments on both simulated and real-world datasets show it reaches new state-of-the-arts results.

Questions:
- I don’t think the ablation is convincing – why does the proportion of non-critical parameters is assumed to be the same as the noise rate? The ablation only shows the method is insensitive when the estimation of the noise rate is not precise. However, why do we need to estimate the noise rate? What if the proportion of non-critical parameters is assumed to be a constant number? In other words, what will the performance be like if the estimation is largely different from the real noise rate?
- Since the validation set is also noisy, why does the early stopping criterion adopts the minimum classification error on it?

---

> ### Author Response · Authors · 2020-11-15
> **The response to Reviewer4**
>
> Thanks for your review and suggestions.
>
> 1. Why does the proportion of non-critical parameters is assumed to be the same as the noise rate?
> - Thank you for your nice concern! We have the intuition that if the label noise rate is high, we will have a large set of incorrect labels and a small set of correct labels. Then, there could be a large number of non-critical parameters which overfit the incorrect labels; while a small number of critical parameters which fit the correct labels. We conjecture that the number of non-critical parameters is proportional to the label noise rate and therefore use the label noise rate to help identify the non-critical parameters. We agree with the reviewer that the non-critical parameters may not rely strictly on the label noise rate, which needs further investigations. For the proposed method, which firstly studies the critical and non-critical parameters, it empirically works well as supported by comprehensive experimental results. We will add similar discussions in the paper to reflect the reviewer’s insight.
>
> 2. What will the performance be like if the proportion of non-critical parameters is assumed to be a constant number or estimation of the noise rate is too bad?
> - Thanks for the suggestion! First of all, as shown in the answer to the above question, it is intuitive to relate the proportion of non-critical parameters to the label noise rate, which is task-dependent and can provide a reliable solution. However, it is interesting to see how it works if we set the proportion of non-critical parameters as a constant number. We would like to mention that it may be hard to set or tune the constant number for the proportion because we only exploit noisy data in this paper. In the paper, we use the noisy validation set for early stopping. Note that the correct labels are dominating in each noisy class and that label noise is random, the accuracy on the noisy validation set and the accuracy on the clean test data set are positively correlated. The noisy validation set therefore can be employed. We will try to use the noisy validation set to locate constants for the proportion of the non-critical parameters on MNIST, F-MNIST, and CIFAR10 in the updated version. Specifically, we will compare the constants and will discuss the relationship between the constants and the label noise rate.
>
> - If the estimation is largely different from the real noise rate, a too large estimation error may hurt the performance of the proposed method. However in fact, the noise rate can be effectively estimated [1-2]. In this paper, we use the estimated noise rate to achieve the superior performance. When the estimation error is within an acceptable range, our method is robust as shown in the ablation study. Your comments about setting the proportion of non-critical parameters are really constructive. We agree that it is an important point needed in-depth study.
>
> 3. Why does the proposed method use a noisy validation set for early stopping?
> - Our work follows the often used practices in the literature of learning from noisy labels, using a noisy validation set for early stopping. The intuition to do so is discussed in the answer to the above question. It empirically works well as also supported by the existing methods, e.g., GCE [3], Forward [4], and T-Revision [5]. A more in-depth theoretical analysis of this aspect is worth further learning.
>
> [1] Tongliang Liu and Dacheng Tao. Classification with noisy labels by importance reweighting. IEEE TPAMI, 2016.\
> [2] Xiyu Yu, Tongliang Liu, Mingming Gong, Kayhan Batmanghelich, and Dacheng Tao. An efficient and provable approach for mixture proportion estimation using linear independence assumption. CVPR, 2018. \
> [3] Zhilu Zhang and Mert Sabuncu. Generalized cross entropy loss for training deep neural networks with noisy labels. NeurIPS, 2018.\
> [4] Giorgio Patrini, Alessandro Rozza, Aditya Krishna Menon, Richard Nock, and Lizhen Qu. Making deep neural networks robust to label noise: A loss correction approach. CVPR, 2017.\
> [5] Xiaobo Xia, Tongliang Liu, Nannan Wang, Bo Han, Chen Gong, Gang Niu, and Masashi Sugiyama. Are anchor points really indispensable in label-noise learning? NeurIPS, 2019.

---

### Official Review · AnonReviewer2 · 2020-10-26
**Review 2 for "Robust early-learning: Hindering the memorization of noisy labels"**

**Rating:** 7
**Confidence:** 4

**Review:**

------Overall------
This paper utilizes the memorization effects of deep models and aims to improve their robustness to noisy labels before early stopping. As the deep models fit training data with clean labels in the early stage of training, the authors propose a novel method to identify those more important parameters for fitting clean labels. They then deactivate the unimportant parameters to reduce side effects brought by noisy labels, which enhances the fitting to clean labels implicitly. I think this paper is interesting and makes sense. The major comments and issues are as follows:

------Major comments------
1. Different from other complex methods for learning with noisy labels , this work discusses that standard cross entropy loss can achieve competitive performance with early stopping. We can therefore focus the training stage before early stopping to handle noisy labels. The authors skillfully allow the optimality to be checked by a scalar, and then judge the importance of the parameters by analyzing the influence of the parameters on this scalar. The idea of this paper is novelty and meaningful.

2. The paper is very well-written. The description of its motivation and technical details is clear and flows smoothly, which makes it easy for readers to understand the core idea of this paper and follow its implementation details.

3. The experimental results are convincing. The authors provide a very detailed description of experimental settings. Besides, this paper exploits multiple methods for comparison and considers various noise settings to verify the effectiveness of the proposed method. The experimental results on synthetic and real-world datasets are convincing. The authors also perform an ablation study to present the proposed method is insensitive to the estimation of noise rate.

------Issues------
1. I only find the illustration of comparison between CE and CDR in the case of noisy CIFAR-100. This paper aims to reduce the side effect of noisy labels before early stopping, thus CE is an importance baseline in this paper. Can the authors add illustrations of the experimental results in other cases like Figure.2?

2. The baselines and experimental results are sufficient. Could the authors add some introduction for the baselines and more detailed discussion for experimental results.

3. The authors may need add some explanation for Eq.(2) and Eq.(5). The proposed method makes use of the memorization effects of deep models. The authors directly write \tilde{S} rather than S in Eq.(5). However, this may be easy to misunderstand. I suggest that the authors can emphasize it or change it.

4. Some typos need to be corrected. (1) “The underlying issue of directly using the gradient of......”; (2) “Robust positive update uses the gradients to update the critical ones......”.

5. Some minor comments. (1) The experimental results in Table 1 are too dense. (2) The figures are not readable, especially the title is small for me. This makes it a little hard to match the figures with specific cases. (3) The parameter (noise rate) $\tau$ still needs to be estimated, which may be challenging. It will be promising to automatically set this parameter during training. Thus, a more in-depth analysis is worthy of further learning.

I hope the authors can address these issues carefully to improve this work.

---

> ### Author Response · Authors · 2020-11-15
> **The response to Reviewer2**
>
> Thank you for your comments! We will answer the questions one by one.
>
> 1. Add illustrations of the experimental results.
> - We agree with you that figures in some cases can better demonstrate the effectiveness of our method. We will add some figures in the supplementary materials. We will keep you informed of the change once done.
>
> 2. Could the authors add descriptions for the baselines and experimental results?
> - Yes. We will update the descriptions accordingly.
>
> 3. The symbol \tidle{S} makes readers misunderstand.
> -  We emphasize it after Eq.(5) to make it clear.
>
> 4. Some typos need to be corrected.
> - Thanks! We have checked it and updated it.
>
> 5. Some minor comments.
> - To increase the readability of the paper, we will take your advice to modify the tables and figures.
> - Yes, it is really worthy of learning. We consider that the AutoML technique used in S2E may be useful for setting the parameter automatically. We will explore in this direction to improve our work in future.

---

### Official Review · AnonReviewer3 · 2020-10-26
**Interesting idea with convincing experimental performance**

**Rating:** 7
**Confidence:** 5

**Review:**

This paper aims to exploit the early stopping method to solve the problem of learning with noisy labels. Specifically, this paper finds that only partial parameters (critical parameters) are important for fitting clean labels and generalize well; while the other parameters (non-critical parameters) tend to fit noisy labels and cannot generalize well. Based on this observation, this paper proposes to divide all parameters into the critical parameters and non-critical ones, and perform different update rules for the two types of parameters, in each iteration. Extensive experiments on benchmark-simulated and real-world label-noise datasets demonstrate the effectiveness of the proposed method.

This paper has the following advantages:
1.	This paper is well-written and the motivation is very clear. This paper has clearly explained the two types of parameters, i.e., critical parameters and non-critical parameters.
2.	This paper proposes a novel method with an interesting idea. Unlike many existing methods that are aggregations of multiple techniques, this paper only focuses on one concept. It is simple but effective. The view of updating different parameters by different rules is quite novel for learning with noisy labels. I think this view may bring some new insights to the area of learning with noisy labels.
3.	Experiments are quite thorough and results on both synthetic and real-world datasets validate the effectiveness of the proposed method.

This paper also has some minor issues:
1.	I would suggest the authors to carefully check the notations used in the paper. For example, $\mathbf{w}$ denotes all the learnable parameters of the model in the paper, while $\mathbf{w}$ usually means a vector. So I would suggest the authors to use another symbol to denote the set of all the learnable parameters of the model, e.g., $\mathcal{W}$.
2.	It is not well justified why this paper chooses the $\ell_1$ regularizer. Is there any consideration to use the $\ell_1$ regularizer except that could be associated with weight decay?
3.	The hyper-parameter $\tau$ (noise rate) needs to be known. It may not be a big issue as the ablation study in this paper demonstrates that the proposed method is insensitive to this hyper-parameter.

Overall, I think this is a good paper with an interesting idea and convincing experimental performance. So I prefer to accept it.

---

> ### Author Response · Authors · 2020-11-15
> **The response to Reviewer3**
>
> Thank you for the valuable feedback! Your suggestions will make this paper better without doubt.
>
> 1. Change the symbol to avoid the confusion.
> - Thank you for this suggestion! We have changed the symbol of the learnable parameters.
>
> 2. The reasons for using the $\ell_1$ regularizer.
> - Thank you for raising this concern. Recall our proposed method and the optimality criterion, the non-critical parameters are penalized to be zero (or close to zero), which makes them to have much less effect on generalization. For the negative update rule in this paper, when we exploit the $\ell_1$ regularizer, it is effective to achieve that non-critical parameters are pushed to be very sparse and be very close to zero. This theory is also support by [1].
>
> 3. The issue about hyper-parameter $\tau$.
> - If the noise rate is unknown, we can estimate it with [2-3]. The methods for estimating noise rate are widely used in existing work, e.g., Co-teaching and Co-teaching+. Additionally, the proposed method is insensitive to the estimation result of the noise rate as you mentioned.
>
> [1] Bishop, Christopher M. Pattern recognition and machine learning. Springer, 2006.\
> [2] Tongliang Liu and Dacheng Tao. Classification with noisy labels by importance reweighting. IEEE TPAMI, 2016.\
> [3] Xiyu Yu, Tongliang Liu, Mingming Gong, Kayhan Batmanghelich, and Dacheng Tao. An efficient and provable approach for mixture proportion estimation using linear independence assumption. CVPR, 2018.

---

### Official Review · AnonReviewer1 · 2020-10-29
**A novel and effective method for learning with noisy labels**

**Rating:** 7
**Confidence:** 4

**Review:**

This paper tackles the problem of learning with noisy labels and proposes a novel method CDR which is inspired by the lottery ticket hypothesis. In particular, the proposed method categorizes the parameters into two parts, including critical parameters and non-critical parameters, and applies different update rules to these parameters. Using comprehensive experiments on synthetic datasets and real-world datasets, the authors verify that the proposed method can improve the robustness of the classifiers against noisy labels.

Pros.
1. The proposed method is interesting in its design. The authors provide an alternative interpretation for the optimality criterion, which reveals that the value of the parameter can also be used to check the optimality. It is novel and of significance. Meanwhile, the proposed method is easy to implement. This method can also be applied to existing algorithms to further improve their robustness.
2. Overall, this paper is very well written and well organized. The technical details are easy to follow, and Algorithm 1 helps understand the procedures.
3. Extensive experiments are performed to verify that the proposed method indeed helps over the baselines at fighting noisy labels. I like the details of experimental settings, which really can help reproduce these experimental results.

Cons.
1. The new interpretation for the optimality criterion is important, but lacks detailed explanation. I find that there is only simple analysis. Perhaps the authors could add some details or citations for better understanding.
2. Though the baseline S2E uses AutoML, it seems that S2E is an improvement on co-teaching or co-teaching+? It is not suitable to place it at the end.
3. For some baselines such GCE and Joint, there are hyperparameters to consider. The authors should explain how to set their values for a fair comparison. For APL, there are multiple combinations of loss functions. In this experiment, which one did you choose? I suggest that the authors add such explanation in Section 4.2, which will make the results more convincing.
4. The proposed method implicitly exploits the memorization effects of deep models, and can reduce the side effect of noisy labels before early stopping. After early stopping, noisy labels still affect the performance. How to reduce the side effect during the whole training by using this idea? I personally think this is an interesting and meaningful direction. The authors can regard this as future work to improve this paper.

Overall, I think this paper makes sense in learning with noisy labels. The proposed method is novel and effective. I recommend to accept this paper, and hope that the authors can address the above issues carefully.

---

> ### Author Response · Authors · 2020-11-15
> **The response to Reviewer1**
>
> Thanks for your constructive suggestions! We have checked and will address the issues as mentioned in “Cons” to improve this paper.
>
> 1. Add details or citations for better understanding the optimality criterion.
> - We will add detailed analyses and explanations for the optimality criterion.
>
> 2. Some modifiabilities in the experiments.
> - We agree with your comment, and place the baseline S2E after Co-teaching+.
> - To make the results more convincing, we add the settings of the baseline methods in Section 4.2.
>
> 3. The advice for future work.
> - Exciting suggestions! It is very interesting and valuable to effectively detect the critical/non-critical parameters during the whole training process, and further achieve a robust classifier. We will investigate this in future work.

---

### Public Comment · ~Ehsan_Amid1 · 2020-11-10
**Please consider referencing/comparing to these more recent works**

I would like to point out that our work (Amid et al. 2019a) extends the Generalized CE loss (Zhang and Sabuncu 2018) by introducing two temperatures t1 and t2 which recovers GCE when t1 = q and t2 = 1. Our more recent work, called the bi-tempered loss (Amid et al. 2019b) extends these methods by introducing a proper (unbiased) generalization of the CE loss and is shown to be extremely effective in reducing the effect of noisy examples. Please consider referencing/comparing to these papers.

(Amid et al. 2019a) Amid et al. "Two-temperature logistic regression based on the Tsallis divergence." In The 22nd International Conference on Artificial Intelligence and Statistics (AISTATS), 2019.

(Amid et al. 2019b) Amid et al. "Robust bi-tempered logistic loss based on Bregman divergences." In Advances in Neural Information Processing Systems (NeurIPS), 2019.

---

> ### Author Response · Authors · 2020-11-15
> **The response to Ehsan Amid**
>
> Hello, Ehsan,
>
> Thank you for your interesting work. We will study them after the rebuttal session.
>
> Best,\
> Paper 1387 authors

---

### Author Response · Authors · 2020-11-24
**Revised draft uploaded**

Dear reviewers and all,

We have revised our draft according to your constructive comments. Major revisions are highlighted in green. We sincerely thank all the reviewers. We would highly appreciate it if you could read our responses and revisions. Please feel free to let us know if further details/explanations would be helpful.

Best,

Authors

---

### Comment · ~Yuchen_Lu1 · 2021-03-29
**Comparisons with ELR**

Hi Authors,

I am wondering how can this paper be compared with ELR? It seems that the CIFAR10 symmetric noise 40% performance is lower than ELR. https://arxiv.org/abs/2007.00151

---

### Decision · Program_Chairs · 2021-01-07
**Final Decision**

**Decision:**

Accept (Poster)

**Comment:**

The paper presents a novel method for learning with noisy labels based on an interesting insight into the learning dynamics of deep neural networks.

Reviewers unanimously vote for acceptance. I agree with their assessment, and it is my pleasure to recommend the paper for acceptance.

If I can draw attention to one comment, I strongly agree with R1 that the criterion in Eq. (3) is somewhat poorly motivated. I believe the paper would benefit from a clearer exposition of this part.

Please make sure to address all reviewers' remarks in the camera-ready version. Thank you for submitting your work to ICLR.

---

> ### Comment · ~Xinshao_Wang1 · 2021-01-18
> **Concern on the OPTIMALITY CRITERION**
>
> Dear authors:
>
> Great work! Your idea on dividing parameters to critical ones and non-critical ones is very interesting for me.
>
> In the section 3.1, G(t) = L(tW; S),  grad_G(1) = grad_L(W;S) * W.tranpose(),
> I find that the OPTIMALITY CRITERION is sufficient, but may be not necessary.
>
> **Sufficient condition**:
> 1. When  grad_L(W;S) = zero vector, we have grad_G(1) = zero scalar.
>
>
> However, **it is not a necessary condition**:
>
> 2. When grad_G(1) = zero scalar, grad_L(W;S) may not be a zero vector.
>
> Therefore, according to my personal understanding so far, the optimality criterion is improper.  Could you kindly explain more to address my concern?
> I am looking forward to your ideas and discussing with you. Please kindly and feel free to correct me if I am wrong.
>
> Many thanks.

---

> > ### Author Response · Authors · 2021-02-06
> > **Response**
> >
> > Dear Xinshao:
> >
> > Thanks for your comments and we are agree with you, i.e., the new optimality criterion is sufficient, but is not necessary. We focus on learning with noisy labels in this paper. The necessity of  the new optimality criterion does not affect the effectiveness of the proposed method.  We will add discussions in the final version to address your concern and enhance this paper.
> >
> > Best,\
> > Authors